# Maximum Average Randomly Sampled: A Scale Free and Non-parametric Algoritm for Stochastic Bandits

**Masoud Moravej Khorasani**[*]
Department of Electrical and Electronic Engineering
The University of Melbourne
Parkville, Melbourne, Victoria, Australia
m.moravej.kh@gmail.com

**Erik Weyer**
Department of Electrical and Electronic Engineering
The University of Melbourne
Parkville, Melbourne, Victoria, Australia
ewey@unimelb.edu.au

## Abstract

Upper Confidence Bound (UCB) methods are one of the most effective methods in dealing with the exploration-exploitation trade-off in online decision-making problems. The confidence bounds utilized in UCB methods tend to be constructed based on concentration equalities which are usually dependent on a parameter of scale (e.g. a bound on the payoffs, a variance, or a subgaussian parameter) that must be known in advance. The necessity of knowing a scale parameter a priori and the fact that the confidence bounds only use the tail information can deteriorate the performance of the UCB methods. Here we propose a data-dependent UCB algorithm called MARS (**M**aximum **A**verage of **R**andomly **S**ampled Rewards) in a non-parametric setup for multi-armed bandits with symmetric rewards. The algorithm does not depend on any scaling, and the data-dependent upper confidence bound is constructed based on the maximum average of randomly sampled rewards inspired by the work of Hartigan in the 1960s and 70s. A regret bound for the multi-armed bandit problem is derived under the same assumptions as for the $\psi$-UCB method without incorporating any correction factors. The method is illustrated and compared with baseline algorithms in numerical experiments.

## 1 Introduction

The classical stochastic multi-armed bandit problem introduced by [21] has been studied extensively in statistics, computer science, electrical engineering, and economics. New applications in machine learning and data science such as personalized new recommendation and Monte-Carlo tree search have generated renewed interest in the problem.

The problem is described as follows: The classical stochastic multi-armed bandit problem is an online decision-making problem where each arm represents an action. At each round $t \in \{1, \cdots, n\}$ the agent chooses action (arm) $A_t \in \mathcal{A}$ and observe a reward $X_t \in \mathbb{R}$ where $\mathcal{A}$ is the set of available actions and its finite cardinality is $K \geq 2$. The reward $X_t$ is a sample from an underlying distribution

---

[*]The code to reproduce the results of the paper can be found at https://github.com/Masoud-Moravej/MARS-NeurIPS2023.

37th Conference on Neural Information Processing Systems (NeurIPS 2023).

$\nu_{A_t}$ with bounded mean $\mu_{A_t}$. Hence, a stochastic bandit is a set of distributions $\nu = \{\nu_i : i \in \mathcal{A}\}$ which is unknown for the agent.

The agent aims to minimize the following regret over $n$ rounds.

$$R_n = n \max_{i \in \mathcal{A}} \mu_i - \mathbb{E}\left[\sum_{t=1}^{n} X_t\right]. \tag{1}$$

with respect to the actions $A_1, \ldots, A_n$.

The minimization of the regret necessitates balancing the trade-off between exploration and exploitation where exploration means investigation to accumulate more information about arms and exploitation means the maximization of the immediate performance. See the book [19] and survey [7] for comprehensive discussion.

A class of algorithms for multi-armed bandit problems that strikes a balance between exploration and exploitation is Upper Confidence Bound (UCB) methods [3, 4, 12, 1].

The UCB algorithm is based on *the principle of optimism in the face of uncertainty*. According to this principle, one should construct probabilistically optimistic guesses for the expected payoff (mean) of all actions and play the arm with the highest guess. The performance of the UCB algorithms in dealing with the exploration-exploitation dilemma depends on the tightness of the upper confidence bound. The literature includes several approaches which construct upper confidence bound based on concentration inequalities such as Hoeffding-type inequalities [4] and self-normalized inequalities [1].

As an example $\psi$-UCB Method introduced in [7] assumes that there exist a convex function $\psi(\cdot)$ which satisfy the following conditions.

$$\forall t, \lambda \geq 0, \quad \log \mathbb{E}\{e^{\lambda(X_t - \mathbb{E}\{X_t\})}\} \leq \psi(\lambda) \text{ and } \log \mathbb{E}\{e^{\lambda(\mathbb{E}\{X_t\} - X_t)}\} \leq \psi(\lambda). \tag{2}$$

Let $\psi^*(\cdot)$ be the Legendre–Fenchel transform of $\psi(\cdot)$, that is

$$\psi^*(\varepsilon) \triangleq \sup_{\lambda \in \mathbb{R}}(\lambda \varepsilon - \psi(\lambda)),$$

Then, with probability at least $1 - \delta$,

$$\hat{\mu}_{i,s} + (\psi^*)^{-1}\left(\frac{1}{s}\log\left(\frac{1}{\delta}\right)\right) > \mu_i$$

where $\hat{\mu}_{i,s}$ is the sample mean of rewards obtained by pulling arm $i$ for $s$ times.

At time $t$, using the obtained UCB and letting $\delta = 1/t^2$, we select

$$A_t = \operatorname{armax}_i\left[\hat{\mu}_{i,T_i(t-1)} + (\psi^*)^{-1}\left(\frac{3\log(t)}{T_i(t-1)}\right)\right]$$

where $T_i(t-1)$ denotes the number of times action $i$ was chosen by the learner after the end of round $t - 1$. It was proved [7] that:

$$R_n \leq \sum_{i:\Delta_i > 0}\left(\frac{3\Delta_i}{\psi^*(\Delta_i/2)}\log(n) + 3\right) \tag{3}$$

where $\Delta_i$ is the suboptimality gap $\Delta_i = \max_{i \in \mathcal{A}} \mu_i - \mu_i$.

However, there are two weaknesses associated with the bounds obtained by concentration inequalities as above. First, they are commonly conservative since they are not data-dependent and only care about tail information of distribution. Second, those bounds usually depend on scaling factors or functions, e.g. $\psi(\cdot)$ in $\psi(\cdot)$-UCB and poor choices can make the bounds conservative.

## 1.1 Related Works

Recently several works have focused on non-parametric bandit algorithms based on subsampling and bootstrapping [5, 6, 18, 17, 22]. Those works use the empirical distribution of the data instead of fitting a given model to the data. The Garbage In, Reward Out (GIRO) method [18] relies on the

history of past observed rewards and enhances its regret bound by augmenting fake samples into the history. The Perturbed-History Exploration method (PHE) [17] serves as a faster and memory-efficient alternative to GIRO. However, PHE has the limitation of being restricted to bounded distributions and involves a tunable parameter. The Residual Bootstrap exploration (ReBoot) [22] perturbs the history in order to improve the regret bound.

Bootstrapping Upper Confidence Bound uses bootstrap to construct sharper confidence intervals in UCB-type algorithm [12]. However, a second-order correction is used to guarantee the non-asymptotic validity of the bootstrap threshold. The second-order correction is not sharp, and it includes scaling factors.

Another line of works which include the Best Empirical Sampled Average method (BESA) [5] and the Sub-sampling Duelling Algorithms (SDA) [6] use subsampling to conduct pairwise comparison (duels) between arms. BESA organize the duels between arms and find the winner by comparing the empirical average of sub-sampled rewards. SDA extends the concept of BESA duels to a round-based comparison by incorporating a sub-sampling scheme and it eliminates the need for forced exploration.

Apart from Reboot which was analysed for Gaussian distributions, and SDA which was analysed for a family of distribution satisfying a balance condition (including Gaussian and Poisson), the other algorithms were analysed for distributions with known bounded support.

## 1.2 Contributions

In this paper, we propose a new non-parametric UCB algorithm which is fully data-dependent without any scale parameter. The proposed approach called MARS (**M**aximum **A**verage **R**andomly **S**ampled) constructs a data-dependent upper confidence bound by selecting the maximum average among randomly sampled rewards. This upper confidence bound is inspired by Leave-out Sign-dominant Correlation Regions (LSCR) method which construct non-asymptotic confidence regions for parameters of dynamical systems [8, 9, 11, 15, 20, 16]. The upper confidence bound inspired by LSCR is *non-asymptotic* and *probabilistically guaranteed*, i.e. it is larger than the true value with an exact user-chosen probability and for a finite number of data points.

The problem-dependent regret bound of the method is given for multi-armed bandits where all underlying distributions $\{\nu_i, i \in \mathcal{A}\}$ are symmetrically distributed about their mean e.g. exponential, uniform, or logistic distribution and inequality (2) is satisfied. As $\psi(\cdot)$ does not need to be known, MARS avoids the problem with sub-optimal choice of $\psi(\cdot)$, and still achieves logarithmic regret.

## 1.3 Notation

A subset of positive integers up to and including a constant $k$ is denoted as $[k] \triangleq \{1, \cdots, k\}$. $\lceil x \rceil$ is the smallest integer greater than or equal to $x$. The cardinality of a set $S$ is denoted by $|S|$, e.g. $|[k]| = k$. $\mathbb{I}\{A \text{ predicate}\}$ is the indicator function defined as

$$\mathbb{I}\{A \text{ predicate}\} = \begin{cases} 1 & \text{if the predicate is TRUE,} \\ 0 & \text{otherwise,} \end{cases}$$

Without loss of generality assume the first arm is optimal i.e. $\mu_1 = \max_i \mu_i$. Define the sub-optimality gap $\Delta_i = \mu_1 - \mu_i$ for $i \in \{2, \cdots, K\}$. Further, let

$$T_i(t) = \sum_{s=1}^{t} \mathbb{I}\{A_s = i\}$$

be the number of times action $i$ was chosen by the learner after the end of round $t$. $X_{i,j}$ is the reward the $j$th time arm $i$ is pulled. $\mathcal{U}(a, b)$ is a uniform distribution bounded by $a$ and $b$ ($a < b$), and $\mathcal{B}(a)$ is a Bernoulli distribution with mean $a$.

The remainder of the paper is organized as follows. In the next section, a new randomized approach to compute an upper confidence bound is presented. In Section 3 the non-parametric UCB algorithm is introduced and a regret bound for the multi-armed bandit problem with symmetric rewards is given. Section 4 explores the numerical performance of the proposed MARS method, and Section 5 concludes the paper.

## 2 A New Randomized Upper Confidence Bound

In this section, an approach to compute a data-dependent upper confidence bound for a finite number of observations is proposed.

Hartigan in [14] proposed an approach which is useful to compute confidence interval for an unknown parameter. The confidence interval was computed by a number of estimates of the mean using different subsets of the data. The estimates of the mean in [12] was computed using a number of balanced sets of subsamples determined by group theory.

The paper [9] used random strings instead of balanced sets of subsamples which brings computational advantage. Now, we use the method in [14] and [9] to propose a new non-asymptotic upper confidence bound.

We have the observations $\{X_{i,s}\}_{s=1}^{T_i}$ from arm $i$ and we want to find an upper confidence bound $\text{UCB}_i$ with probability $1 - \delta$ such that

$$\Pr\{\text{UCB}_i(T_i, \delta) \geq \mu_i\} \geq 1 - \delta$$

In this section $T_i$ is assumed to be a fixed number of observations which does not depend on other actions taken or observed rewards. In Section 3, the situation where $T_i$ is dependent on the past actions and observed rewards will be considered.

Before introducing the new upper confidence bound, let us define typical values.

**Definition 1.** *The set of ascending-ordered and continuously distributed random variables $Z_{(1)}, Z_{(2)}, \ldots, Z_{(M-1)}$ is a set of typical values for $\mu$ if the probabilities that $\mu$ belongs to each of following intervals are the same*

$$\left(-\infty, Z_{(1)}\right), \left(Z_{(1)}, Z_{(2)}\right), \ldots, \left(Z_{(M-1)}, \infty\right),$$

*that is*

$$\Pr\left\{\mu \in \left(-\infty, Z_{(1)}\right)\right\} = \Pr\left\{\mu \in \left(Z_{(1)}, Z_{(2)}\right)\right\} = \cdots = \Pr\left\{\mu \in \left(Z_{(M-1)}, \infty\right)\right\} = \frac{1}{M}.$$

Now, based on Theorem 1 in [9] the following theorem introduces a set of typical values for the means of $i$th arm $\mu_i$.

**Theorem 1.** *We have the observations $\{X_{i,s}\}_{s=1}^{T_i}$ from arm $i$ and $\mathbb{E}\{X_{i,s}\} = \mu_i$ for all $s \in [T_i]$. Assume that the observations $\{X_{i,s}\}$ admit continuous distributions and are independent and symmetrically distributed about their mean.*

*Then, a set of typical values for $\mu_i$ is given by*

$$\hat{\mu}_i^j = \frac{1}{\sum_{s=1}^{T_i} h_{j,s}} \sum_{s=1}^{T_i} h_{j,s} X_{i,s} \text{ for } j \in [M - 1] \tag{4}$$

*where $M \leq 2^{T_i}$ is a positive integer and $\{h_{j,t}\}$ are independent random sequences such that*

$$\Pr\{h_{j,s} = 0\} = \Pr\{h_{j,s} = 1\} = \frac{1}{2}.$$

*A string $\{h_{j,s}\}_{s=1}^N$ is discarded if it turns out to be equal to an already constructed string.*

*Proof.* Follows along the same line as the Proof of Theorem 1 in [9]. See Section 1 in the supplement for further details. □

Theorem 1 shows that $M - 1$ estimates of mean computed by random sub-sampling are a set of typical values for $\mu_i$ and partition the real line into equiprobable segments where $\mu_i$ belong to each one of those segments with equal probability. As a consequence of Theorem 1 we conclude that

$$\Pr\{\max_{j \in [M-1]} \hat{\mu}_i^j > \mu_i\} = 1 - 1/M.$$

$M$ is a user-chosen parameter and hence the probability of the confidence bound can be tuned by the user. Since $M \leq 2^{T_i}$ the maximum confidence which can be achieved by computing subsample

$$\text{UCB}_i(T_i, \delta) = \begin{cases} \max_s X_{i,s} & \text{w.p. } \delta 2^{T_i} \\ \infty & \text{w.p. } 1 - \delta 2^{T_i} \\ \max_j \hat{\mu}_i^j(t), & \text{otherwise} \end{cases}, \quad \text{if } T_i < \frac{\log(\delta^{-1})}{\log(2)}$$

Table 1: Data-dependent and scale-free UCB.

means is $1 - 1/2^{T_i}$ and we can not get an upper confidence bound with arbitrary high confidence. To overcome this problem, we use the fact that all means $\mu_i$ are bounded and the following upper confidence bound is proposed.

Compute $M = \lceil \delta^{-1} \rceil$ random subsample means as in (4). The non-parametric upper confidence bound is given in Table 1.

The probability of the constructed upper confidence bound is presented in the next theorem.

**Theorem 2.** *Let $T_i$ be a fixed number of observations from arm $i$. For each $i \in [K]$ assume $\{X_{i,s}\}_{s=1}^{T_i}$ are independent random variables with continuous distribution functions which are symmetrically distributed about its mean $\mu_i$. Let the user-chosen parameter $\delta \in (0,1)$ be chosen such that $\delta^{-1}$ is an integer. Then for all $T_i > 0$ the following non-asymptotic equality holds*

$$\Pr\{UCB_i(T_i, \delta) \geq \mu_i\} = 1 - \delta \tag{5}$$

*Note that $T_i$ is deterministic, and the probability is with respect to both $\{X_{i,s}\}$ and $\{h_{j,s}\}$.*

*Proof.* See Section 2 in the supplement. $\square$

Theorem 2 shows that the probability of the proposed confidence region is exact and hence it is not conservative. This result holds for a finite number of observations as long as the reward is symmetrically distributed around its mean and $T_i$ is a fixed given number.

**Example 1** In Figure 1, we compare different approaches to calculate 95% UCB for the population mean based on samples from a Gaussian and an uniform distribution. The upper confidence bounds are calculated for different number of data points. The almost exact quantile is computed by finding 95% quantile of the population of sample means available by repeating experiment 10000 times. Naive bootstrap threshold and bootstrapped threshold were computed by equation (2.6) in Remark 2.3 in [12].

The figures shows that for most sample sizes the MARS bound is sharper than naive bootstrapped threshold, bootstrapped threshold, sub-Gaussian bound, and Hoeffding bound. The naive bootstrap is sharper for small sample sizes, but as shown in [12], naive bootstrap do not produce reliable results for small samples sizes. It is also notable that the MARS bound is probabilistically guaranteed for a finite number of data points as shown in Theorem 2.

## 3 Algorithm and Regret Bound

The UCB method using the proposed upper confidence bound for $n$ rounds is given in Algorithm 1. At the initial rounds, the algorithm tends to adopt an optimistic outlook in the face of uncertainty. This is reflected in the upper confidence bound, which is set to infinity or the highest reward observed. As the number of rounds increases, the upper confidence bounds approach the mean reward for each arm with high probability. Consequently, the algorithm progressively leans towards exploitation.

If two or more UCBs in line 15 of Algorithm 1 take on the same value, randomly choose an action among the minimizers.

MARS necessitates keeping $\lceil 1/\delta \rceil$ sub-sampled means for each arm. This leads to a memory requirement of $O(Kn)$ when $\delta = 1/n$. The computational complexity of MARS is also depends

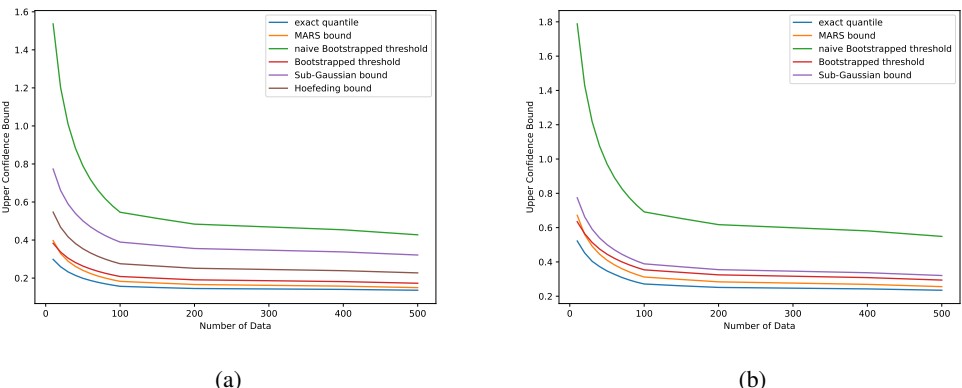

(a)                                                                 (b)

Figure 1: 95% UCBs of sample means computed by different approaches. The underlying distributions are uniform between $-1$ and $+1$ in (a) and Gaussian with mean $0$ and variance $1$ in (b). The results are averaged over 10000 realisations.

---

**Algorithm 1 M**aximum **A**verage of **R**andomly **S**ampled (MARS)

---

**Input:** $\delta$

1:  $M \leftarrow \lceil \delta^{-1} \rceil$                                                                                    $\triangleright$ Initialization

2:  $\forall i \in [K], \ \forall j \in [M] : T_i(0) \leftarrow 0, H_{j,i} \leftarrow 0$

3:  **for** $t \leftarrow 1$ to $n$ **do**

4:     **for** $i \leftarrow 1$ to $K$ **do**                                        $\triangleright$ Computing upper confidence bound

5:         **if** $T_i(t-1) = 0$ **then**

6:             $\text{UCB}_i(T_i(t-1), \delta) \leftarrow \infty$

7:         **else if** $0 < T_i(t-1) < \frac{\log(\delta^{-1})}{\log(2)}$ **then**

8:             Draw $x \sim \mathcal{U}(0,1)$

9:             **if** $x < \delta 2^{T_i(t-1)}$ **then**

10:                 $\text{UCB}_i(T_i(t-1), \delta) \leftarrow \infty$

11:             **else**

12:                 $\text{UCB}_i(T_i(t-1), \delta) \leftarrow \max_s X_{i,s}$

13:         **else**

14:             $\text{UCB}_i(T_i(t-1), \delta) \leftarrow \max_j \hat{\mu}_i^j(t)$

15:     $A_t \leftarrow \operatorname{argmax}_i \text{UCB}_i(T_i(t-1), \delta)$                             $\triangleright$ Pulled arm

16:     Pull Arm $A_t$ and observe reward $X_t$

17:     **for** $i \leftarrow 1$ to $K$ **do**                                            $\triangleright$ Update statistics

18:         **if** $i = A_t$ **then**

19:             $T_i(t) \leftarrow T_i(t-1) + 1$

20:             **for** $j \leftarrow 1$ to $M$ **do**

21:                 Draw $h_{j,i} \sim \mathcal{B}(0.5)$

22:                 $H_{j,i} \leftarrow H_{j,i} + h_{j,i}$

23:                 $\hat{\mu}_i^j(t) \leftarrow \frac{(H_{j,i-1})\hat{\mu}_i^j(t-1)+h_{j,i}X_t}{H_{j,i}}$

24:         **else**

25:             **for** $j \leftarrow 1$ to $M$ **do**

26:                 $T_i(t) \leftarrow T_i(t-1)$

27:                 $\hat{\mu}_i^j(t) \leftarrow \hat{\mu}_i^j(t-1)$

---

on the choice of $\delta$, as updating $\lceil 1/\delta \rceil$ sub-sampled means is performed in each round. Notably, to reduce the computational burden, the required Bernoulli variables in the algorithm can be generated and stored before the start of the game.

The Next theorem provides a regret bound for the proposed MARS algorithm when applied to multi-armed bandit problems with symmetric distributions.

**Theorem 3.** *Let $T_i(n) > 0$ be the total number of observation from arm $i$ and $\delta = 1/n^2$. The following assumptions about all arms $i \in [K]$ are in place.*

- *$\{X_{i,s}\}_{s=1}^{T_i(n)}$ are continuously distributed and independent random variables symmetrically distributed about its mean $\mu_i$. The observations $X_{i,s}$ for each arm $i \in [K]$ are independent of the observations from other arms and actions.*

- *For all $i \in [K]$ and $s \in [T_i(n)]$, there is a convex function $\psi_i(\lambda)$ such that:*

$$\log \mathbb{E}\{e^{\lambda(X_{i,s}-\mathbb{E}\{X_{i,s}\})}\} \leq \psi_i(\lambda)$$

*Then the regret is bounded by*

$$R(n) \leq \sum_{i:\Delta_i>0} \left(3 + \frac{3\log(n)}{x_i}\right)\Delta_i, \tag{6}$$

*where*

$$x_i = -\log\left(\frac{1}{2} + \frac{1}{2}\exp(-\psi_i^*(\Delta_i))\right),$$

*and $\psi_i^*(\cdot)$ is the Legendre–Fenchel transform of $\psi_i(\cdot)$, that is*

$$\psi_i^*(\varepsilon) \triangleq \sup_{\lambda \in \mathbb{R}}(\lambda\varepsilon - \psi_i(\lambda)).$$

*Note that the expectation in the regret is with respect to both $\{X_{i,s}\}$ and $\{h_{j,s}\}$.*

*Proof.* See Section 3 in the supplement. □

In Theorem 3 we set $\delta = 1/n^2$ to simplify analysis. A similar regret bound can also be achieved using $\delta_t = 1/t^2$. It is also notable that the regret bound in Theorem 3 was achieved without any knowledge about the functions $\psi_i(\lambda)$ apart from its existence. In particular, it was not used when the upper confidence bounds were computed. However, the $\psi$-UCB method assumed that $\psi_i(\lambda)$s are known and used it in constructing the upper confidence bound.

The next corollary characterises a relationship between the regret bounds of the current method and those of the $\psi$-UCB method.

**Corollary 1.** *Provided that $0 < \psi_i^*(\Delta_i) < 1.59$, the regret bound (6) is simplified as follows*

$$R(n) \leq \sum_{i:\Delta_i>0} \left(3 + \frac{6\log(n)}{\psi_i^*(\Delta_i)}\right)\Delta_i. \tag{7}$$

*Now, let for all arms $i \in [K]$, the corresponding reward is $\sigma_i$-sub Gaussian. Then, $\psi_i^*(\Delta_i) = (\Delta_i^2)/(2\sigma_i^2)$. Accordingly when $(\Delta_i^2)/(2\sigma_i^2) < 1.59$ the regret bound is simplified to*

$$R(n) \leq \sum_{i:\Delta_i>0} \left(3\Delta_i + \frac{12\sigma_i\log(n)}{\Delta_i}\right). \tag{8}$$

*Proof.* See Section 4 in the supplement. □

As shown in equation (6), the regret bound for MARS is always $O(\log(n))$ without relying on the use of $\psi(\cdot)$. This demonstrates the effectiveness of the introduced non-parametric UCB method. When $\psi_i^*(\Delta_i) < 1.59$, the task becomes more challenging as identifying the optimal arms becomes harder. In such scenarios, both the regret bounds for the proposed MARS and the $\psi$-UCB, which employs $\psi(\cdot)$, become dependent on the function $\log(n)/\psi_i^*(\Delta_i)$. Corollary 1 also explore the effectiveness of MARS when dealing with subgaussian rewards. It demonstrates that even without prior knowledge of the $\sigma_i$ values, MARS successfully addresses bandit problems, achieving a regret bound of $O(\sum_{i:\Delta_i>0} \log(n)/\Delta_i)$ for challenging scenarios where $(\Delta_i^2)/(2\sigma_i^2) < 1.59$.

# 4 Experiments

In this section, the performance of the proposed MARS method is compared with upper confidence bound based on concentration inequalities (Vanilla UCB, See e.g. Chapter 7 and 8 in [19]), Thompson sampling with normal prior (Normal-TS) [2], Bootstrapped UCB [12], Garbage In, Reward Out (GIRO) [18], and Perturbed-History Exploration (PHE) [17]. The package PymaBandits[2] was used to implement these baseline methods.

The probability of confidence regions in the Vanilla UCB, Bootstrapped UCB, and the proposed MARS UCB is set to $0.999$, i.e. $\delta = 1/1000$. In GIRO, the parameter $a$, which represents pseudo-rewards per unit of history, is set to $1$. Due to the high sensitivity of the PHE approach to the choice of the tunable parameter $a$, simulations were performed for two values of $a$.

The number of arms is $K = 5$ and the means are

$$\mu_1 = 1, \ \mu_2 = \frac{1}{2}, \ \mu_3 = \frac{1}{3}, \ \mu_4 = \frac{1}{4}, \ \mu_5 = \frac{1}{5}.$$

First, consider the case where the rewards are Gaussian with variance $1$ for all arms. The cumulative regrets are show in Figure 2a. Since Normal-TS uses distribution knowledge and the variances are correct it is the best as expected. The Vanilla UCB algorithm demonstrates comparable or superior performance compared to both the Bootstrapped UCB, MARS, and GIRO. The performance of the PHE approach is heavily dependent on the parameter $a$. When $a = 2.1$, it shows a linear regret. However, for $a = 5.1$, it outperforms most other approaches, except for Thompson sampling.

Both Vanilla UCB and Normal-TS depend on the variances which were assumed known in the previous simulation. We repeat the simulation with the variances incorrectly set to $2$. The result is shown in Figure 2b. Evidently MARS, GIRO, and Bootstrapped UCB outperform both Vanilla UCB and Normal-TS when incorrect values for the variances are used. MARS demonstrates superior performance over Bootstrapped UCB and GIRO after an initial set of rounds. Moreover, unlike GIRO and Bootstrapped UCB, MARS does not require the full storage of reward history, resulting in lower computational complexity. As previously observed, PHE with a value of $2.1$ demonstrated the poorest performance, whereas PHE with a value of $5.1$ has the best performance.

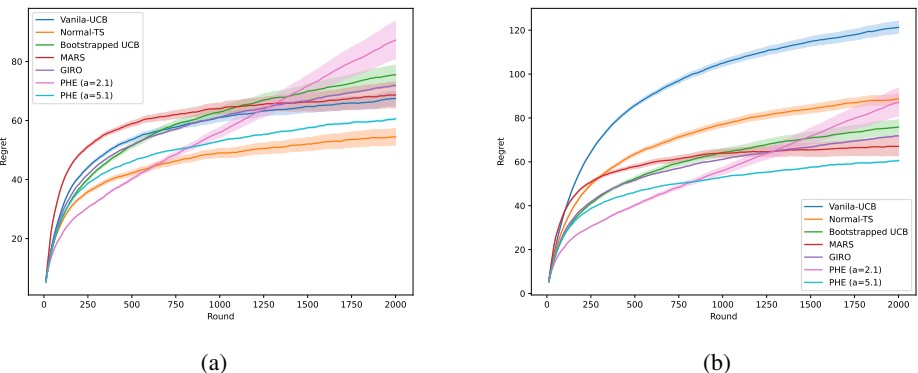

(a)  (b)

Figure 2: Cumulative regret for Gaussian bandit. The variances of rewards are true in (a) and wrong in (b). The results are averaged over 2000 realizations.

The MARS and GIRO do not use the tail information of the rewards. However, Vanilla UCB, Normal-TS, and Bootstrapped UCB use the distribution and the tail information of the rewards respectively and their performance can deteriorate when the prior knowledge is wrong or conservative. To illustrate this we repeated the simulation for the cases where the rewards admit uniform distribution over $[-1, 1]$. The results are shown in Figure 3. It shows that MARS which does not use the distribution of the rewards and the tail information, outperforms the other methods except PHE ($a = 2.1$) in this case. An interesting observation is that PHE ($a = 5.1$) exhibits outstanding performance in a Gaussian setup, yet it performs poorly in a Uniform setup, indicating a strong reliance on the tunable

---
[2] The package can be found at `https://www.di.ens.fr/olivier.cappe/Code/PymaBandits`.

parameter. This dependence on the parameter could pose challenges in real-world applications where the environment is unknown. In such cases methods like MARS and GIRO may be more practical alternatives.

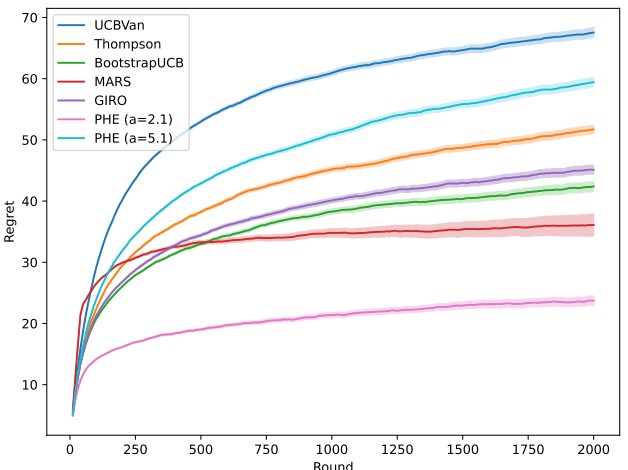

Figure 3: Cumulative regret for uniform bandit. The results are averaged over 2000 realizations.

See Section 5 in the supplement for further simulations.

## 5 Conclusion

In this paper, a non-parametric and scale-free confidence bound was proposed based on random sub-sampling of rewards. The confidence bound was used to propose a data-driven UCB algorithm. A regret bound of the method for multi-armed bandit problems with symmetric distribution was presented, and it was shown that the regret bound is always $O(\log(n))$. Moreover, the regret bound is $O(\sum_{i:\Delta_i>0} \log(n)/\Delta_i)$ for a class of Bandit problems. Numerical experiments show that the proposed method performs well, and it outperforms baselines algorithms when the prior information is mis-specified.

Extending the method to other bandit and reinforcement learning problems are interesting topics for future works e.g. evaluating the method on non-stationary bandits and contextual bandits using the ideas in [13] and [9]. As MARS does not use any concentration inequalities in the construction of the confidence bound, the application of MARS to bandits with heavy tail is another interesting direction. Another interesting future direction is evaluation of the method on asymmetric rewads and robustification of the approach following ideas in [10].

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
