# Supplement to "Maximum Average Randomly Sampled: A Scale Free and Non-parametric Algorithm for Stochastic Bandits"

Masoud Moravej Khorasani and Erik Weyer

Department of Electrical and Electronic Engineering
The University of Melbourne, Victoria 3010, Australia
m.moravej.kh@gmail.com, ewey@unimelb.edu.au

## Contents

## 1   Proof of Theorem 1

The following lemma given in [2] is useful for the proof of Theorem 1.

**Lemma 1.** *[2] Given a stochastic matrix*

$$H = \begin{bmatrix} 0 & 0 & \cdots & 0 \\ h_{1,1} & h_{1,2} & \cdots & h_{1,T_i} \\ \vdots & & & \vdots \\ h_{M-1,1} & h_{M-1,2} & \cdots & h_{M-1,T_i} \end{bmatrix}$$

*where $\{h_{i,s}\}$ is introduced in Theorem 1. Let $\eta = [\eta_1, \cdots, \eta_{T_i}]^\top$ be a vector independent of $H$ of mutually independent random variables symmetrically distributed about 0. Fix a $\bar{j} \in [0, M]$, the matrix $H_{\bar{j}}$ is defined as the $M \times T_i$ matrix whose rows are all equal to the $\bar{j}$th row of $H$. Then, $H\eta$ and $(H - H_{\bar{j}})\eta$ have the same M-dimensional distribution provided that the $\bar{j}$th element of $(H - H_{\bar{j}})\eta$ is repositioned as first element of the vector.*

Now, let $\eta = [\eta_1, \cdots, \eta_{T_i}]^\top = [X_{i,1} - \mu_i, \ X_{i,2} - \mu_i, \ \cdots, \ X_{i,T_i} - \mu_i]^\top$ and define $\zeta \triangleq [\zeta_0, \cdots, \zeta_{M-1}]^\top = H\eta$. We know that $\zeta_0 = 0$ base on the definition of the matrix $H$.

Pick a variable $\zeta_{\bar{j}}$. $\zeta_{\bar{j}}$ is in the $r$th position if the inequality $\zeta_{\bar{j}} > \zeta_j$ (or $\zeta_j - \zeta_{\bar{j}} < 0$) holds for exactly $r - 1$ choices of $j \in \{0, \cdots, M - 1\}$.

$\zeta_j - \zeta_{\bar{j}}$ can be rewritten as

$$\zeta_j - \zeta_{\bar{j}} = \sum_{s=1}^{T_i} h_{j,s}\eta_s - \sum_{s=1}^{T_i} h_{\bar{j},s}\eta_s = \sum_{s=1}^{T_i}(h_{j,s} - h_{\bar{j},s})\eta_s$$

Hence, the condition $\zeta_{\bar{j}} > \zeta_j$ holding for $r - 1$ choices of $j$ is the same as the condition that $r - 1$ entries of $(H - H_{\bar{j}})\eta$ are negative. From Lemma 1 we have that $(H - H_{\bar{j}})\eta$ and $H\eta$ have the same distribution. Therefore, the probability of the events that $r - 1$ entries of $(H - H_{\bar{j}})\eta$ and $H\eta$ are negative are the same. But the event that $r - 1$ entries of $H\eta$ are negative does not depend on $\bar{j}$, showing that the probability of being in the $r$th position is the same for any $\zeta_{\bar{j}}$. Since $\bar{j}$ can take on $M$ possible values, this probability is $1/M$, and hence

$$\Pr\{\zeta_0 = 0 \text{ is in the } r\text{th position}\} = \frac{1}{M}, \text{ for all } r \in [M]$$

Now, for all $r \in [M]$ we have

$$\Pr\{\zeta_0 = 0 \text{ is in the } r\text{th position}\} = \Pr\{0 > \zeta_j \text{ holds true for } r - 1 \text{ choices of } j\}$$

$$= \Pr\left\{0 > \sum_{s=1}^{T_i} h_{j,s}\eta_s \text{ holds true for } r - 1 \text{ choices of } j\right\}$$

$$= \Pr\left\{0 > \sum_{s=1}^{T_i} h_{j,s}(X_{i,s} - \mu_i) \text{ holds true for } r - 1 \text{ choices of } j\right\}$$

$$= \Pr\left\{\mu_i > \frac{\sum_{s=1}^{T_i} h_{j,s}X_{i,s}}{\sum_{s=1}^{T_i} h_{j,s}} \text{ holds true for } r - 1 \text{ choices of } j\right\}$$

$$= \Pr\{\mu_i > \hat{\mu}_i^j \text{ holds true for } r - 1 \text{ choices of } j\} = \frac{1}{M},$$

which shows that $\{\hat{\mu}_i^j\}$ $j = 1 \ldots, M - 1$ is a set of typical values for $\mu_i$.

## 2   Proof of Theorem 2

The following propositions are used to prove this theorem.

**Proposition 1.** *Suppose $\{y_s\}_{s=1}^N$ are independent and continuously distributed random variables symmetrically distributed about its mean $\mu$. Then*

$$\Pr\{\max_s y_s > \mu\} = 1 - \frac{1}{2^N}$$

*Proof.* We have $\Pr\{\max_s y_s \leq \mu\} = \Pr\{\text{All } y_s \leq \mu\}$ since if there was any $s$ such that $y_s > \mu$

$$\max_s y_s \geq y_s > \mu$$

Since $y_s$ are independent and continuously distributed random variables we have

$$\Pr\{\text{All } y_s \leq \mu\} = \prod_{i=1}^N \Pr\{y_i < \mu\} = \left(\frac{1}{2}\right)^N$$

$\square$

**Proposition 2.** *Assume $\{y_s\}_{s=1}^N$ are continuously distributed and independent random variables symmetrically distributed about its mean $\mu$. Define $M - 1$ random subsample means as follows:*

$$\hat{\mu}_i^j(N) = \frac{1}{\sum_{s=1}^N h_{j,s}} \sum_{s=1}^N h_{j,s} y_s \text{ for } j \in [M - 1],$$

*where $\{h_{j,t}\}$ are independent random sequences such that*

$$\Pr\{h_{j,s} = 0\} = \Pr\{h_{j,s} = 1\} = \frac{1}{2}.$$

*and a string $\{h_{j,s}\}_{s=1}^N$ is discarded if it turns out to be equal to an already constructed string.*

*The upper confidence bound is given by*

$$UCB_i(N, \delta) = \max_j \hat{\mu}_i^j$$

*has the following non-asymptotic property*

$$\Pr\{UCB_i(N, \delta) \geq \mu_i\} = 1 - \frac{1}{M}$$

*Proof.* Along the same line as the proof of Theorem 1 it can be shown that $\{\hat{\mu}_i^j(N)\}_{j=1}^{M-1}$ is a set of typical values for $\mu$. Hence,

$$\Pr\{\max_j \hat{\mu}_i^j \geq \mu_i\} = 1 - \frac{1}{M}$$

$\square$

$\delta^{-1}$ is chosen as an integer, so $M = \delta^{-1}$. If $T_i \geq \frac{\log(\delta^{-1})}{\log(2)}$ (equivalent $2^{T_i} \geq \delta^{-1}$) using Proposition 2 we have

$$\Pr\{\max_j \hat{\mu}_i^j(t) > \mu_i\} = 1 - \frac{1}{M} = 1 - \delta.$$

Now, consider the case $0 < T_i < \frac{\log(\delta^{-1})}{\log(2)}$ (or $1 < 2^{T_i} < \delta^{-1}$). In this case, there is not enough observations to achieve an upper confidence bound using Proposition 2. The randomized UCB for this case has also an exact confidence as illustrated below:

$$
\begin{aligned}
&\Pr\{\text{UCB}_i(T_i, \delta) > \mu_i\} \\
=&\Pr\{\text{UCB}_i(T_i, \delta) > \mu_i | \text{UCB}_i(T_i, \delta) = \infty\} \Pr\{\text{UCB}_i(T_i, \delta) = \infty\} \\
&+ \Pr\{\text{UCB}_i(T_i, \delta) > \mu_i | \text{UCB}_i(T_i, \delta) \neq \infty\} \Pr\{\text{UCB}_i(T_i, \delta) \neq \infty\} \\
=&\Pr\{\infty > \mu_i\}(1 - \delta 2^{T_i}) + \Pr\{\max_s X_{i,s} > \mu_i\}(\delta 2^{T_i}) \\
=&(1 - \delta 2^{T_i}) + (1 - \frac{1}{2^{T_i}})\delta 2^{T_i} = 1 - \delta.
\end{aligned}
$$

In the second equality, the boundedness of the means of the arms and Proposition 1 were utilized.

# 3    Proof of Theorem 3

The steps in this proof closely follows the proof of Theorem 7.1 in [3].

Consider the following decomposition for the regret

$$R_n = \sum_{i=1}^{K} \Delta_i \mathbb{E}[T_i(n)]. \tag{1}$$

To upper bound the regret, we bound the expected number of pulls $\mathbb{E}[T_i(n)]$ for sub-optimal arms. Let us define a 'good' event as

$$G_i = \left\{ \mu_1 < \min_{t \in [n]} \text{UCB}_1(T_i(t), \delta) \right\} \cap \{\text{UCB}_i(u_i, \delta) < \mu_1\}$$

where $u_i \in [n]$ is a deterministic value chosen later. The event $G_i$ happens where we never underestimate the UCB of the optimal arm (arm 1) and at the same time the upper confidence bound of sub-optimal arm $i$ after $u_i$ observations is less than the means of optimal arm.

We are going to show

1. If $G_i$ occurs $T_i(n) \leq u_i$

2. By choosing a proper $u_i$ the complement event $G_i^c$ occurs with low probability

Thanks to the mentioned points which will be shown we have

$$\mathbb{E}[T_i(n)] = \mathbb{E}[\mathbb{I}\{G_i\}T_i(n)] + \mathbb{E}[\mathbb{I}\{G_i^c\}T_i(n)] \leq u_i + \mathbb{P}(G_i^c)n \tag{2}$$

In the latter inequality we used the fact that $T_i(n) \leq n$.

Here (2) is shown. Let us first show $T_i(n) \leq u_i$ when the event $G_i$ occurs. By contradiction suppose $T_i(n) > u_i$ where the event $G_i$ occurs. Then arm $i$ was played more than $u_i$ times and there must exist a round $t \in [n]$ such that $T_i(t-1) = u_i$ and $A_t = i$. However, the definition of event $G_i$ implies

$$\text{UCB}_i(t, \delta) = \text{UCB}_i(u_i, \delta)$$
$$< \mu_1 < \text{UCB}_1(t, \delta)$$

which means $A_t = \text{argmax}_j \text{UCB}_j(t-1) \neq i$. It is a contradiction and we have $\mathbb{E}[\mathbb{I}\{G_i\}T_i(n)] \leq u_i$.

Now, $\mathbb{P}(G_i^c)$ is evaluated and it is shown that it is small. By the definition of $G_i$ we have

$$G_i^c = \left\{ \mu_1 \geq \min_{t \in [n]} \text{UCB}_1(t, \delta) \right\} \cup \{\text{UCB}_i(u_i, \delta) \geq \mu_1\} \tag{3}$$

The first set can be included in the following set by union trick.

$$\left\{ \mu_1 \geq \min_{t \in [n]} \text{UCB}_1(t, \delta) \right\} \subset \bigcup_{s \in [n]} \{\mu_1 \geq \text{UCB}_1(s, \delta)\}$$

Then using Theorem 2 we obtain:

$$\mathbb{P}\left( \mu_1 \geq \min_{t \in [n]} \text{UCB}_1(t, \delta) \right) \leq \mathbb{P}\left( \bigcup_{s \in [n]} \{\mu_1 \geq \text{UCB}_1(s, \delta)\} \right)$$
$$\leq \sum_{s=1}^{n} \mathbb{P}\left( \mu_1 \geq \text{UCB}_1(s, \delta) \right) = n\delta \tag{4}$$

Notice that $s$ the values 1 to $n$ that $s$ can take in (4) are deterministic.

The next step is to bound the probability of the second set in (3). First we introduce a lemma which is useful in finding that bound.

**Lemma 2.** *Suppose $\{y_s\}_{s=1}^{N}$ are independent random variables with mean $\mu$ and there exist a convex function $\psi(\lambda)$ such that for all $s$ and $\lambda > 0$*

$$\log \mathbb{E}\{e^{\lambda(y_s - \mathbb{E}\{y_s\})}\} \leq \psi(\lambda) \text{ and } \log \mathbb{E}\{e^{\lambda(\mathbb{E}\{y_s\} - y_s)}\} \leq \psi(\lambda).$$

$\{h_s\}_{s=1}^{N}$ *are independent random variables such that*

$$\Pr\{h_j = 0\} = \Pr\{h_j = 1\} = \frac{1}{2}.$$

*Then,*

$$\mathbb{P}\left( \frac{1}{\sum_{j=1}^{N} h_j} \sum_{s=1}^{N} h_s y_s - \mu > \varepsilon \right) \leq \exp(-xN),$$

*where*

$$x = -\log\left( \frac{1}{2} + \frac{1}{2}\exp(-\psi^*(\varepsilon)) \right),$$

*and $\psi^*(\cdot)$ be the Legendre–Fenchel transform of $\psi(\cdot)$.*

*Proof.* Using Markov's inequality we have

$$\mathbb{P}\left(\frac{1}{N}\sum_{s=1}^{N} y_s - \mu > \varepsilon\right) \le \exp(-\psi^*(\varepsilon)N), \tag{5}$$

In this lemma, we use $\omega$ to show the dependence of some empirical variables on the observed realization. Let $H(\omega) \triangleq \sum_{j=1}^{N} h_j$.

$$\mathbb{P}\left(\frac{1}{\sum_{j=1}^{N} h_j}\sum_{s=1}^{N} h_s y_s - \mu > \varepsilon\right)$$

$$= \sum_{J=0}^{N} \mathbb{P}\left(\frac{1}{H(\omega)}\sum_{s=1}^{H(\omega)} y_s - \mu > \varepsilon \Big| H(\omega) = J\right)\mathbb{P}\left(H(\omega) = J\right) \quad \text{(The law of total probability)}$$

$$\le \sum_{J=0}^{N} \exp(-J\psi^*(\varepsilon))\binom{N}{J}\left(\frac{1}{2}\right)^J\left(\frac{1}{2}\right)^{N-J} \quad \begin{array}{l}\text{((5) and } H(\omega) \text{ has a}\\\text{binomial distribution)}\end{array}$$

$$= \sum_{J=0}^{N} \binom{N}{J}\left(\frac{1}{2}\exp(-\psi^*(\varepsilon))\right)^J\left(\frac{1}{2}\right)^{N-J}$$

$$= \left(\frac{1}{2} + \frac{1}{2}\exp(-\psi^*(\varepsilon))\right)^N = \exp(-xN)$$

$\square$

We return to upper bound the probability of the second set in (3). Assume that $u_i$ is chosen large enough such that

$$u_i \ge \frac{\log(\delta^{-1})}{\log(2)} = \frac{2\log(n)}{\log(2)}. \tag{6}$$

Then the following equality holds

$$\text{UCB}_i(u_i, \delta) = \max_j \hat{\mu}_i^j(u_i),$$

and we have

$$\mathbb{P}\left(\text{UCB}_i(u_i, \delta) \ge \mu_1\right) = \mathbb{P}\left(\max_j \hat{\mu}_i^j(u_i) \ge \mu_1\right) \le \sum_{j=1}^{M-1} \mathbb{P}\left(\hat{\mu}_i^j(u_i) \ge \mu_1\right)$$

$$\le \sum_{j=1}^{M-1} \mathbb{P}\left(\hat{\mu}_i^j(u_i) - \mu_i \ge \Delta_i\right) \le M\exp(-x_i u_i) \tag{7}$$

where

$$x_i = -\log\left(\frac{1}{2} + \frac{1}{2}\exp(-\psi_i^*(\Delta_i))\right).$$

The last inequality was found using Lemma 2. Note that $u_i$ is deterministic in this setting.

Now, using (4), (7), and the choice $\delta = 1/n^2$ the inequality (2) is rewritten as

$$\mathbb{E}[T_i(n)] \leq u_i + n^2 \delta + nM \exp(-x_i u_i)$$
$$\leq u_i + 1 + n^3 \exp(-x_i u_i)$$

We need to chose $u_i$ such that (6) holds. Considering the fact that $x_i < 0.7$ by its definition we set

$$u_i = \left\lceil \frac{3 \log(n)}{x_i} \right\rceil > \frac{2 \log(n)}{\log(2)}.$$

By substituting $u_i$ the regret (1) is rewritten as

$$R_n = \sum_{i:\Delta_i > 0} \Delta_i (u_i + 1 + n^3 \exp(-x_i u_i)) \leq \sum_{i:\Delta_i > 0} \Delta_i \left( \left\lceil \frac{3 \log(n)}{x_i} \right\rceil + 1 + n^3/n^3 \right)$$
$$\leq \sum_{i:\Delta_i > 0} \Delta_i \left( 3 + \frac{\log(n^3)}{x_i} \right)$$

# 4  Proof of Corollary 1

The aim is to find a lower bound for $x_i = -\log\left(\frac{1}{2} + \frac{1}{2}\exp(-\psi_i^*(\Delta_i))\right)$. Since for all $Z > 0$, $\log(Z) \leq Z - 1$ we have

$$x_i = -\log\left(\frac{1}{2} + \frac{1}{2}\exp(-\psi_i^*(\Delta_i))\right) \geq \frac{1}{2} - \frac{1}{2}\exp(-\psi_i^*(\Delta_i))$$

For all $Z \in [0 \ 1.59]$ the inequality $\exp(-Z) \leq 1 - Z/2$ holds. Hence, $\exp(-\psi_i^*(\Delta_i)) \leq 1 - \psi_i^*(\Delta_i)/2$. The lower bound for $x_i$ is simplified as follows:

$$x_i \geq \frac{1}{2} - \frac{1}{2}\exp(-\psi_i^*(\Delta_i)) \geq \frac{1}{2} - \frac{1}{2} + \frac{\psi_i^*(\Delta_i)}{2} \geq \frac{\psi_i^*(\Delta_i)}{2}. \tag{8}$$

Using the lower bound for $x_i$ we have

$$R(n) \leq \sum_{i:\Delta_i > 0} \left( 3 + \frac{3 \log(n)}{x_i} \right) \Delta_i \leq \sum_{i:\Delta_i > 0} \left( 3 + \frac{6 \log(n)}{\psi_i^*(\Delta_i)} \right) \Delta_i,$$

where $0 < \psi_i^*(\Delta_i) < 1.59$. For sub Gaussian pay off $\psi_i^*(\Delta_i) = (\Delta_i^2)/(2\sigma_i^2)$ and substituting this yields (8) in the main paper.

# 5  Numerical Experiments

Here, we present additional numerical experiments showing the performance of MARS compared to other approaches.

| Method | Run time (seconds) |
|---|---|
| UCB Vanilla | 2.44 |
| Thompson | 3.82 |
| PHE | 5.51 |
| MARS | 23.48 |
| GIRO | 149.15 |

Table 1: Runtimes of algorithms on uniform bandit.

## 5.1 Computational Complexity of MARS

To assess the computational complexity of MARS in comparison to other methods, we present the runtime of all approaches used in the experiment shown in Figure 3 in the paper.

Table 1 displays the average runtime of different approaches for the multi-armed bandit with a Uniform setup when the number of rounds is set to 2000. In MARS, the parameter $\delta$ is set to 1/1000, requiring updates of 1000 subsampled means in each round. As a result, it takes more time compared to Vanilla UCB, Thompson Sampling, and PHE. However, MARS is faster than approaches like GIRO, which store the full memory of rewards.

## 5.2 Reward with Truncated Gaussian Distribution

The numerical experiment conducted in Section 3 of the main paper is replicated in Figure 1, utilizing Gaussian rewards truncated within the range of $[-1, 1]$.

In line with the uniformly distributed rewards presented in the main paper, the results shows that MARS outperforms the other methods except PHE ($a = 2.1$) thanks to not utilizing reward distribution and tail information.

## 5.3 Reward with Exponential Distribution

MARS is a well-suited method for handling a wide range of symmetrically distributed rewards, as it does not rely on tail information. In this context, we replicate the numerical experiment from Section 3 of the main paper, using exponentially distributed rewards. In this experiment, we also include a comparison of MARS with the BESA approach using sub-sampling [1], to provide a comprehensive evaluation.

Figure 2 clearly demonstrates that MARS outperforms the other methods including BESA, PHE ($a = 2.1$), and PHE ($a = 5.1$).

## 5.4 Reward with Bernoulli (non-symmetrical) Distribution

To evaluate the robustness of MARS and its effectiveness where the assumption of symmetric reward distributions is not met, MARS is implemented on a Bernoulli setup where

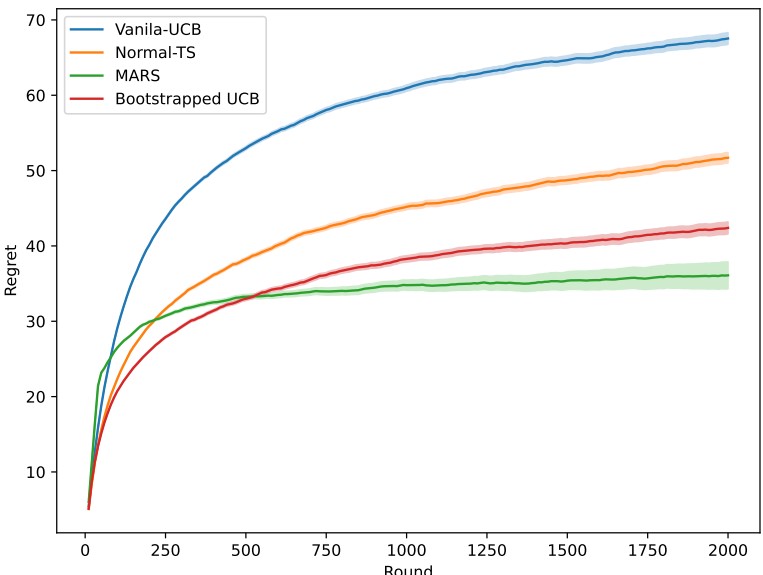

Figure 1: Cumulative regret for truncated-Gaussian bandit. The results are averaged over 2000 realizations.

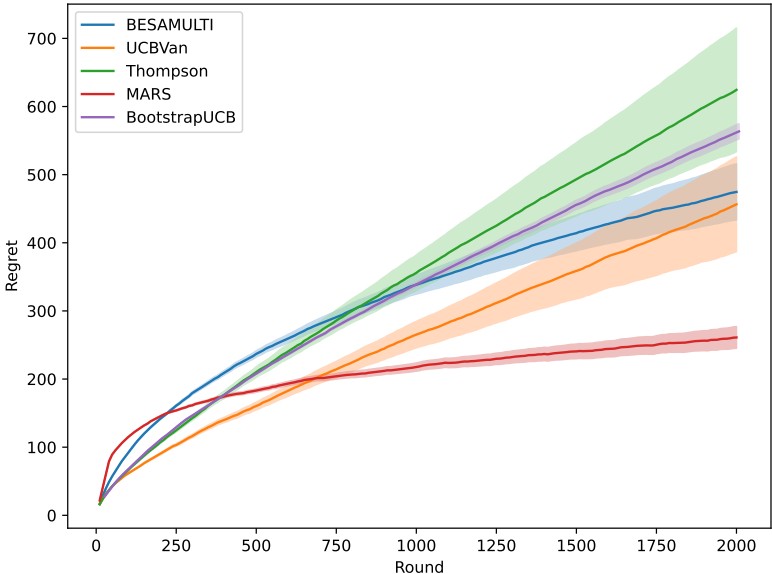

Figure 2: Cumulative regret for exponential bandit. The results are averaged over 2000 realizations.

the number of arms is $K = 2$ and the means are $\mu_1 = 0.5$ and $\mu_2 = 0.01$. The findings are shown in Figure 3. The results indicate that although the setup does not meet the

symmetric assumption, MARS outperforms Vanilla UCB and Thompson sampling. Its performance is also comparable to that of BESA.

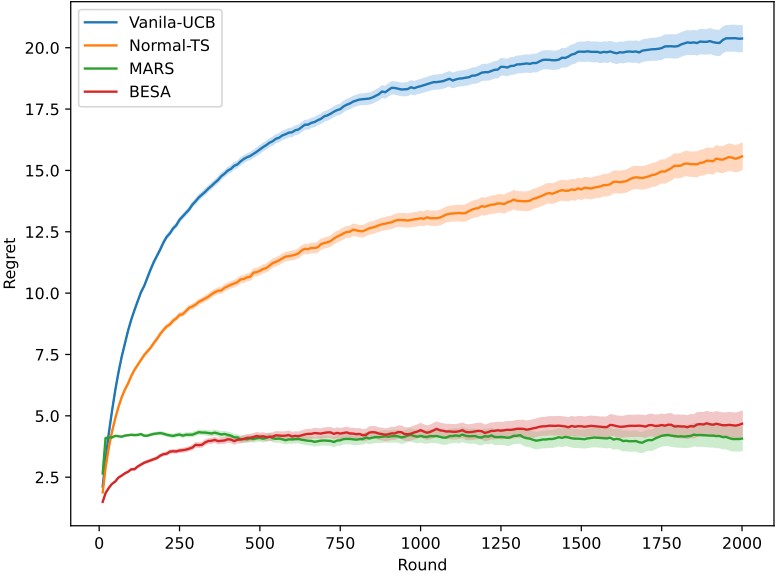

Figure 3: Cumulative regret for Bernoulli bandit. The results are averaged over 2000 realizations.