# OpenReview forum: "Maximum Average Randomly Sampled: A Scale Free and Non-parametric Algorithm for Stochastic Bandits"
_NeurIPS.cc/2023/Conference — NeurIPS 2023 poster_

### Official Review · Reviewer_t4Bj · 2023-06-15

**Soundness:** 4 excellent
**Presentation:** 3 good
**Contribution:** 3 good
**Rating:** 6
**Confidence:** 5

**Summary:**

The authors propose a novel non-parametric algorithm for stochastic bandits, based on a sub-sampling scheme. While other algorithms using sub-sampling have been recently proposed, their approach significantly differs from these works. Indeed, instead of using sub-sampling to perform pairwise comparisons, they use it to build exact confidence intervals used in a UCB algorithm called MARS. To do that, they build on results provided by statisticians in the 60s and 70s, showing that sub-sampling can be used to build a set of *typical values* for the true mean of each distribution.

After describing their method, the authors provide the theoretical guarantees of MARS: logarithmic regret under the assumption that the distributions are continuous and symmetric, and some exponential moment condition similar to the one used for $\psi$-UCB. Interestingly, these guarantees are achieved while the algorithm does not use the function $\phi$ directly, hence avoiding sub-optimal choices of $\psi$, which is the advantage of non-parametric algorithms.

Then, a set of experiments aim at validating the method.

**Strengths:**

I think that the paper is overall well-written and clear, and that the algorithm and theoretical results are well presented and easy to understand.

While the literature in bandits is now quite vast, it is difficult to come up with new principles, especially in the standard stochastic case. Recently, the literature on non-parametric algorithms based on bootstrap and sub-sampling has proposed interesting new approaches, and this paper is in my opinion a nice addition to this literature, showing new potentialities. In particular, the theoretical guarantees hold under original assumptions, for which no other algorithms is proved to work (additional assumptions would be needed). Furthermore, the empirical results validate the approach.

Overall, I think that the paper brings enough interesting insights for publication at Neurips, up to some changes listed below. The derivation of the technical results seem correct to me.

**Weaknesses:**

In my opinion the paper has two major weaknesses, that are related:

* I believe that in the context of the paper the literature review on non-parametric algorithms should be more precise, and the contribution of MARS compared to these approaches should be better explained. First, the presented algorithms are not all based on "subsampling", GIRO, PHE, Bootstrap-UCB and Reboot are closer to bootstrap, while only BESA and SDA are technically speaking based on sub-sampling. Those approaches are rather different, work under different assumptions, and do not share the same pros and cons.

Furthermore, the last line "One drawback of..." is misleading, and make the reader believe that the said drawbacks are shared by all these algorithms, would be solved by MARS, and are in fact the motivation for MARS; while it seems to me that this all these perceived messages are wrong.

In my opinion, there are several axis that should be detailed when talking about these algorithms: the family of distributions for which they work (and maybe how tight are their guarantees for these families), what they need to know about the distributions, and their cost (memory and time). For instance, PHE is a faster and memory-less alternative to GIRO, but its guarantees are sub-optimal and restricted to bounded distributions, and there is a tunable parameter (but this method easily generalize to structured settings, which is its main strength). On the other hand, SDA works in much broader setting (families of distributions for which a "balance" condition is satisfied), but this generality costs the fact of storing all observations in memory: this is in my opinion similar to what is obtained with MARS. Furthermore, the question of improving the computation time and memory for SDA has been studied for the LB-SDA algorithm (http://proceedings.mlr.press/v139/baudry21b/baudry21b.pdf).

* (Related) the memory and computation time of MARS are not well discussed. In particular, it seems worse than most existing approaches: maintaining the sub-sample means for each arm costs $O(Kn^2)$ (if $\delta=1/n^2$ is set), while at most $O(n)$ is needed for the benchmarks. Furthermore, the update of the UCB also requires to sample $n^2$ Bernoulli variables at each time step, which is also much larger the cost of (even the less "optimized") existing non-parametric approach. Those are major drawbacks in my opinion, that should be explicitly discussed, providing comparisons with benchmarks. However, this does not mean than the approach is not interesting: I believe that the authors should really focus on the fact that their algorithms have guarantees under original assumptions for which there are no competitors (or no tight competitors).

* The contribution of the paper is not on the technical side, most results and techniques used by the authors are known, but I think that it is interesting enough so that it is not an issue.

* I find the set of experiments not very convincing: only a few curves/benchmark algorithms are used, and I don't understand why $\delta$ is not set to the theoretically valid threshold ($4\times 10^{-6}$). In my opinion SDA (for instance LB-SDA, which is the fastest SDA) should be present in the benchmarks since it is theoretically valid for all the settings considered. I would like to see a comparison of computation times too.

I believe the authors left enough space so that all these discussions could be added. In my opinion they should be carefully addressed in the revision, and I would be happy to further raise my score in that case.

**Questions:**

* The statistical validity of the estimator holds for large enough sample size. For smaller sample sizes, the author build a UCB using the *maximum of observed data and a fixed probability of returning $\infty$ to keep the $\delta$ confidence. At first read, this step is a bit confusing, and It may have been easier to simply provide the required $\log(n)/\log(2)$ samples per arm at the initialization of the algorithm. Is there an empirical motivation for not doing that (because from the theoretical point of view this do not seem to change much)?
* Related question: is it clear that the $2^{-T_i(n)}$ is tight? Since it is critical in terms of computation/memory cost it may be interesting to optimize it.
* It would be interesting to understand how good the theoretical results are. You started to do that by comparing them with $\psi$-UCB and the usual bound with $\Delta^2$. Did you try to look at the Burnetas & Katehakis lower bound for this specific family of distributions?
* Did you investigate an anytime version of MARS? It seems to me that the proof may easily adapt, making the algorithm more flexible.

**Limitations:**

No potential societal impact.

---

> ### Author Rebuttal · Authors · 2023-08-08
>
> We thank your detailed and insightful review.
>
> **@R4-A1) More precise literature review and clarification of MARS’s contribution**
>
> To enhance the precision and accuracy of the literature review, we revised the "Related Work" section in the paper. The text previously present in lines 56-69 was replaced with the following content:
>
> *“Recently several works have been focused on non-parametric bandit algorithms based on subsampling and Bootstrapping [5,6,16,15,20]. Those works use the empirical distribution of the data instead of fitting a given model to the data.
> GIRO relies on the history of past observed rewards and enhances its regret bound by augmenting fake samples into the history [16]. PHE serves as a faster and memory-efficient alternative to GIRO, demonstrating adaptability to structured settings. However, PHE has the limitation of being restricted to bounded distributions and involves a tunable parameter [15]. Reboot [20] perturbs the history in order to improve the regret bound.*
>
> *Bootstrapping Upper Confidence Bound is using bootstrap to construct sharper confidence intervals in UCB-type algorithm [11]. However, it used a second-order correction to guarantee the non-asymptotic validity of the bootstrap threshold. The second-order correction is not sharp, and it includes some scaling factors.*
>
> *Another line of works including BESA [5] and SDA [6]  use subsampling to conduct pairwise comparison (duels) between arms. BESA organize the duels between arms and find the winner by comparing the empirical average of sub-sampled rewards.  SDA extends the concept of BESA duels to a round-based structure by incorporating a sub-sampling scheme and it eliminates the need for forced exploration.*
>
> *Apart from Reboot which was analysed for Gaussian distributions, and SDA which was analysed for a family of distribution satisfying a balance condition (namely Gaussian and Poisson), the other algorithms were analysed for distributions with known bounded support.”*
>
> The content in line 83 of the contribution section was substituted with the following information to provide a clearer explanation of MARS's contribution.
>
> *“MARS achieves logarithmic regret without using the function $\psi(\cdot)$. Hence it avoids sub-optimal choices of $\psi(\cdot)$”*
>
> **@R4-A2) Memory and Computational Complexity of MARS**
>
> We agreed with the reviewers' observations concerning the computational complexity and memory aspects of the MARS. In order to address this important aspect, we added a paragraph after line 162. Please see Review 1’s Rebuttal section labelled **@R1-A2** for the paragraph and further details.
>
> We conducted an analysis of its runtime with alternative approaches. The relevant table can be found in the attached PDF, which has been added as an addition to Section 5 of the supplementary material. Please see **@R1-A2** for the added text explaining the result of the table.
>
> **@R4-A3) Assessing the Quality of Theoretical Results and Examination of the Burnetas & Katehakis Lower Bound**
>
> The current version presents the logarithmic upper bound for regret without directly utilizing $\psi(\cdot)$. We find the idea of exploring the possibility of establishing a lower bound for regret to be another interesting path for further theoretical investigation of MARS. We sincerely appreciate your valuable suggestion.
>
> **@R4-A4) Necessity of constructing a UCB using either the maximum of observed reward or infinity and possibility of alternative approach providing the necessary log(n)/log(2) samples per arm during the initialization phase**
>
> The mentioned modification enables obtaining a guaranteed confidence bound for any number of observations without directly using $\psi(\cdot)$ function or making any distributional assumptions on arms. The confidence bound in Table 1 was proven to be guaranteed in Theorem 2. Then, the guaranteed bound was used in proving Theorem 3. In equation (4) in supplementary material we have
> $$\mathbb{P}\left(\mu_1\geq{\text{UCB}}_1(s,\delta)\right)= \delta,$$ which implied by Theorem 2.
> Providing log(n)/log(2) samples per arm does not guarantee a confidence region for all values of $\delta$ within the range (0,1).
>
> It is important to note that Table 1 contains a typo, regarding the placement of probabilities in front of "w.p." The probabilities was swapped. The probability of assigning infinity to the upper confidence bound is not constant; rather, it diminishes as the number of times arm $i$ is pulled ($T_i$) increases.
>
> **@R4-A5) Is $2^{T_i(n)}$ tight?**
>
> We think the initial stage of the algorithm is inevitable since we aim to avoid using $\psi(\cdot)$ or any tail information. MARS is in this phase for each arm when $T_t$ is less than $\log(\delta-1)/\log(2)$. For example, when \delta=10^-6, the initial phase occurs for $T_i<2\log(10^6)/\log(2)=39.87$.
>
> It is important to note that we demonstrated that this approach yields logarithmic regret with the proposed initial phase.
>
> **@R4-A6) empirical evaluation and numerical experiments**
>
> Please see Review 3’s Rebuttal, sections labelled **@R3-A1, @R3-A2, @R3-A4**.
>
> **@R4-A7) why $\delta$ is not set to the theoretically valid threshold $4×10^{-6}$**
>
> The guidance provided in the book [17] on pages 104 proposes the following:
>
> _“This suggests we might choose $\delta \approx 1/n$ …”_
>
> As a result, we choose a $\delta=1/1000$. To be fair, the selection of $\delta$ for both Vanilla UCB and Bootstrap UCB is the same.
>
> **@R4-A8) Anytime version of MARS**
>
> Along the same line of Theorem 2.1's proof on Page 11 of reference [7], we can derive a logarithmic upper bound for regret by selecting $\delta_t = 1/{t^2}$.
>
> We would like to thank once again for the time, effort and your insightful feedback.

---

> > ### Comment · Reviewer_t4Bj · 2023-08-11
> > **post-rebuttal comment**
> >
> > I read the other reviews and the authors rebuttal, that confirm my evaluation of the paper. I want to thank the authors for their careful clarification, and I believe that if the authors take in account the comments made in all the reviews for their revision (such as discussing more the practical aspect of the algorithm: memory and empirical performance) the paper will be ready for publication.

---

### Official Review · Reviewer_GPeu · 2023-06-26

**Soundness:** 3 good
**Presentation:** 3 good
**Contribution:** 2 fair
**Rating:** 6
**Confidence:** 4

**Summary:**

The paper presents a new approach to develop a data-dependent upper confidence bound to replace the classical UCB based on concentration inequalities. The data-dependent bound is constructed using sub-sampling of rewards and offers a tighter estimate on the error than the classical UCB resulting in improved performance.

**Strengths:**

The approach adopted in the paper is quite interesting and seems to be significantly simpler than existing studies on data-dependent bounds. Although Theorem 1 is based on an existing result, its application to bandits is interesting and I haven't seen that before.

**Weaknesses:**

I am convinced that there is some novelty in the paper, especially on the theoretical aspect. However, I feel that it may not be sufficient for a paper like this to warrant a publication. While data-dependent bounds are interesting to study and analyze, they themselves have little to offer in terms of advancing the theoretical understanding of multi-armed bandits. These collection of studies that focus on data-dependent bounds are, at a fundamental level, studies that focus on improving practical implementation of MAB. Specifically, as pointed out even by the authors, the classical UCB requires some additional information, that is often difficult to obtain in practice, thereby making such data-dependent approaches a practical alternative.

For a result that is fundamentally based on improving practical implementation, the paper has very limited empirical evaluation. I strongly suggest that the authors add more detailed empirical evaluation, both in terms of comparison with more algorithms like GIRO and PHE and in terms of more complicated examples, preferably a real world example.

A more extensive empirical evaluation will demonstrate the actual improvement offered in practice by this new data-dependent estimator and the existing theoretical results will _support_ and _explain_ those results, making a strong overall paper.

I am willing to increase my score if the authors can add more detailed experiments with proper analysis. I do understand that the rebuttal period might not be sufficient for that in which case, I suggest the authors to resubmit with more experiments. The results definitely seem promising.

EDIT: Score updated based on rebuttal.

**Questions:**

See above.

**Limitations:**

Yes.

---

> ### Author Rebuttal · Authors · 2023-08-08
>
> We appreciate your recognition of the theoretical novelty presented in our work. To address your concerns regarding the empirical evaluation of MARS several experiments were added to the paper described below:
>
> __@R3-A1) Enhancing Empirical Evaluation of MARS in Paper__
>
> As proposed GIRO and PHE were implemented and compared with MARS in all setups, i.e., Gaussian,Truncated Gaussian, Uniform, Exponential, and Bernouli. The specifics of these changes are outlined below.
> The attached PDF file contains updated versions of Figures 2(a), 2(b), and 3. The original Figures 2(a), 2(b), and 3 were replaced with these new versions. After adding the new simulations and experiments, we revised the Section 3 and lines 194 to 220 were replaced with the following text.
>
> *“First, consider the rewards corresponding to all arms are Gaussian with variance 1. The cumulative regrets are show in Figure 1 (a). Since Normal-TS uses the distribution knowledge and the variances are correct in this case it is the best as expected. The Vanilla UCB algorithm demonstrates comparable or superior performance compared to both the Bootstrapped UCB, MARS, and GIRO. The performance of the PHE approach is heavily dependent on the parameter $a$. When $a=2.1$, it shows a linear regret. However, for $a=5.1$, it outperforms most other approaches, except for Thompson sampling.*
>
> *Both Vanilla UCB and Normal-TS depend on the variances which were assumed known in the previous simulation. We repeat the former simulation where the variances are incorrectly set to 2. The result is shown in Figure 2(b). Evidently MARS, GIRO, and Bootstrapped UCB outperform both Vanilla UCB and Normal-TS when incorrect values for variances are used. MARS demonstrates superior performance over Bootstrapped UCB and GIRO after an initial set of rounds. Moreover, unlike GIRO and Bootstrapped UCB, MARS does not require the full storage of reward history, resulting in lower computational complexity. As previously observed, PHE with a value of 2.1 demonstrated the poorest performance, whereas PHE with a value of 5.1 has the best performance.*
>
> *The MARS and GIRO does not use the tail information of the rewards. However, Vanilla UCB, Normal-TS, and Bootstrapped UCB use the distribution and the tail information of the rewards respectively and their performance can deteriorate when the prior knowledge is wrong or conservative.*
>
> *To illustrate this we repeated the simulation for the cases where the rewards admit uniform distribution over $[-1, 1]$. The results are shown in Figure 3. It shows that MARS which does not use the distribution of the rewards and the tail information, outperforms the other methods except PHE ($a=2.1$) in this case. An intriguing observation is that PHE ($a=5.1$) exhibits outstanding performance in a Gaussian setup, yet it performs poorly in a Uniform setup, indicating a strong reliance on the tunable parameter. This dependence on the parameter could pose challenges in real-world applications where the environment is unknown. Consequently, making methods like MARS and GIRO more practical alternatives.*
>
> *For additional simulations in the exponential setup and Gaussian setup, refer to Section 5 in the supplement.”*
>
> __@R3-A2) Enhancing Empirical Evaluation of MARS in Supplementary material__
>
> Three simulations were included in section 5 of supplementary material. The attached PDF file contains revised versions of Figures 1 and 2 in the supplementary. A new simulation and table were added to the supplementary to assess MARS for non-symmetric distributions and compare its runtime with other approaches. Detailed explanations of these additional simulations are provided herein.
>
> __(Truncated Gaussian Setup):__  Methods GIRO, PHE were added to the simulation and the explanation is changed as below
>
> *“The numerical experiment conducted in Section 3 of the main paper is replicated in Figure 1, utilizing Gaussian rewards truncated within the range of $[-1, 1]$. In line with the uniformly distributed rewards presented in the main paper, the results shows that MARS outperforms the other methods except PHE (a=2.1) thanks to not utilizing reward distribution and tail information.”*
>
> **(Exponential Setup)** New methods, GIRO, PHE was added to the simulation, leading to modifications in the explanations.
>
> *“MARS is a well-suited method for handling a wide range of symmetrically distributed rewards, as it does not rely on tail information. In this context, we replicate the numerical experiment from Section 3 of the main paper, using exponentially distributed rewards.
> In this experiment, we also include a comparison of MARS with the BESA approach using sub-sampling [5], to provide a comprehensive evaluation. Figure 2 clearly demonstrates that MARS outperforms the other methods including BESA, PHE (a=2.1), and PHE (5.1).”*
>
> __@R3-A3) Runtime Analysis of the MARS__
>
> To further exploration of the efficiency of MARS and its applicability in real-world scenarios, we conducted an analysis of its runtime with alternative approaches. Please see **@R1-A2** or further details.
>
> __@R3-A4) MARS performance for non-symmetric reward distributions (Bernoulli)__
>
>  The Bernoulli Bandit is a framework for tackling various real-world challenges, like online advertising, where advertisers must make prompt decisions on which ad to present to a user to maximize the likelihood of the user clicking on the ad.
>
> In this scenario, the assumption of MARS, i.e., rewards are distributed symmetrically around the mean, is not satisfied. To evaluate the robustness of MARS and its effectiveness in real-world applications when not all assumptions hold true, we implement MARS in the this setup. Through this evaluation, we can gain insights into MARS's suitability and performance in practical applications despite deviations from ideal assumptions. Please see **@R1-A1** for further details.
>
> We would like to thank once again for the time, effort and your insightful feedback.

---

> > ### Comment · Reviewer_GPeu · 2023-08-14
> > **Response to the authors**
> >
> > Thank you for your detailed response. I feel the experimental results are promising and support the paper well. The authors have addressed my main concern and I am raising my score to 6.

---

### Official Review · Reviewer_DQ4N · 2023-07-04

**Soundness:** 3 good
**Presentation:** 3 good
**Contribution:** 3 good
**Rating:** 6
**Confidence:** 4

**Summary:**

The paper proposed a new method to compute the upper confidence bound (UCB). The new method does not require knowing or estimating the scale parameters. The property shown by Theorem 2 addresses the conservative issue of the existing methods. The practical importance of free from modeling the scale parameters is verified in the experiments.

**Strengths:**

- (a) The paper contributes a new way to compute UCBs of practical importance; it does not require scale information.
- (b) Rigorous theoretical analysis.
- (c) Experiments demonstrate the advantage of the proposed UCBs.

**Weaknesses:**

- (d) What are typical values? It is defined in Definition 1, but its intuition and usage are unclear. Also, it would be great to explain the typical value's role in proving Theorem 2 (the only mention I saw of it is in the proof of Proposition 2).
- (e) Maybe adding a discussion (after Line 192) about recovering the worst-case bound would make the paper more complete?
- (f) Is it better to start the paper with an Introduction?

**Questions:**

- (g) The symmetric assumption restricts the applicability of this submission. Can we remove it? What would the challenge(s) be?
- (h) Visualizing the differences between the choice of scale parameters (Figure 1) and the choice of prior (Figure 2) is nice. Is it possible to demonstrate the difference between the conservative approaches (Lines 51--54) and the non-conservative (Lines 141--143) UCBs?
- (i) Between the scale-free (this submission) and the scale-aware (baselines compared in this submission) approaches, another approach would be estimating the unknown scale parameters. If some scale-estimation approach does not require the symmetric assumption, it might be a strong baseline to be compared. A related paper would be (http://proceedings.mlr.press/v119/zhu20d/zhu20d.pdf).

**Limitations:**

Yes.

---

> ### Author Rebuttal · Authors · 2023-08-08
>
> We thank reviewer for insightful comments and positive feedback on the conducted theoretical analysis.
>
> **@R2-A1) What are typical values? It is defined in Definition 1, but its intuition and usage are unclear. Also, it would be great to explain the typical value's role in proving Theorem 2 (the only mention I saw of it is in the proof of Proposition 2).**
>
> The typical values in Definitions 1 consist of random variables that partition the real line into equally probable segments, where the true mean is positioned within one of these segments with identical probability. Theorem 1 proves that the mean calculated from sub-samples (with a probability of $1/2$) forms a set of typical values. This concept helps to establish a guaranteed upper confidence bound, as shown by Theorem 2 without using any concentration inequalities which are often conservative and including scaling parameters. Typical values is crucial in the proof of theorem 2 which was later used in algorithm and regret analysis.
>
> To clarify that point, the following sentence is included after Theorem 1.
>
> *“Theorem 1 shows that $M-1$ estimates of mean computed by random sub-sampling are a set of typical values for $\mu_i$ and partition the real line into equiprobable segments where $\mu_i$ belong to each one of those segments with equal probability.”*
>
> **@R2-A2) Maybe adding a discussion (after Line 192) about recovering the worst-case bound would make the paper more complete?**
>
> To clarify the regret bound lines 187-192 is revised as below
>
> *“As shown in equation (6), the regret bound for MARS is always $O(\log(n))$ without relying on the use of $\psi(\cdot)$. When $\psi_i^\*(\Delta_i)<1.59$, the task becomes more challenging as identifying the optimal arms becomes harder. In such scenarios, both the regret bounds for the proposed MARS and the $\psi$-UCB, which employs $\psi(\cdot)$, become dependent on the function ${\log(n)}/{\psi_i^\*(\cdot)}$. This demonstrates the effectiveness of the introduced non-parametric UCB method.
> Corollary 1 also explore the effectiveness of MARS when dealing with subgaussian rewards. It demonstrates that even without prior knowledge of the $\sigma_i$ values, MARS successfully addresses bandit problems, achieving a regret bound of $O(\sum_{i:\Delta_i>0}\log(n)/\Delta_i)$  for challenging scenarios where $(\Delta_i^2)/(2\sigma_i^2)<1.59$.”*
>
> **@R2-A3) Is it better to start the paper with an Introduction?**
>
> The lines 17 to 96  of the paper  was  contained in a section labelled Introduction as proposed.
>
> **@R2-A4) The symmetric assumption restricts the applicability of this submission. Can we remove it? What would the challenge(s) be?**
>
> Please see the first response in reviewer 1’s rebuttal labelled **@R1-A1**
>
> **@R2-A5) Visualizing the differences between the choice of scale parameters (Figure 1) and the choice of prior (Figure 2) is nice. Is it possible to demonstrate the difference between the conservative approaches (Lines 51--54) and the non-conservative (Lines 141--143) UCBs?**
>
> In Section 3-Experiment a comparison among MARS using a non-conservative UCB and other methods were done. Two more methods GIRO and PHE were also added to the simulation which are available in the figure in PDF attached.
>
> The Vanila UCB in those simulations uses a concentration inequality including a scaling parameter.
> The Thompson sampling implemented in that section uses Gaussian prior. The performance of these methods was assessed and compared with that of MARS, along with several others.
>
> **@R2-A6) Between the scale-free (this submission) and the scale-aware (baselines compared in this submission) approaches, another approach would be estimating the unknown scale parameters. If some scale-estimation approach does not require the symmetric assumption, it might be a strong baseline to be compared. A related paper would be (mlr.press/v119/zhu20d/zhu20d.pdf).**
>
> We appreciate your suggestion regarding the comparison of MARS with methods that estimate unknown scale parameters. In line with your concerns about enhancing empirical evaluation and conducting comparative analyses with alternative approaches, we added additional simulations into the paper.
>
> As an example, we conducted a comparison of MARS with six alternative approaches within a multi-armed bandit setup with exponential distribution. The outcomes of this comparison are presented in “Figure 3 in Supplement” of the Supplementary section, available in the attached PDF. Among these six approaches, BESA (referred to as BESAMULTI) is a scale-free algorithm that selects arms through duels. As illustrated in the figure, MARS demonstrates superior performance over BESA within this exponential configuration.
>
> For additional empirical assessments and comparisons, please see response labelled as  **@R3-A1** in Rebuttal for Reviewer 3.
>
> We would like to thank once again for your insightful feedback.

---

> > ### Comment · Reviewer_DQ4N · 2023-08-15
> >
> > The feedback clarifies all my questions and provides insights from other reviews' comments. I want to thank the authors' feedback on all reviews and all reviewers' comments. As a result, I would like to keep my original decision.

---

### Official Review · Reviewer_sKFK · 2023-07-04

**Soundness:** 3 good
**Presentation:** 3 good
**Contribution:** 3 good
**Rating:** 7
**Confidence:** 4

**Summary:**

In multi-armed bandits (MAB), when the noise distribution is known, the UCB algorithm with a carefully constructed confidence bound achieves a gap-dependent regret depending on the noise distribution. This manuscript studies the following question: when the noise distribution is unknown, is there an algorithm that adaptively constructs the confidence bound and achieves a near-optimal regret? To answer this question, the authors used a cute observation in [Campi and Weyer, 2010]: if the distribution of a random variable is symmetric around its mean, then the sample means after subsampling (with probability 1/2) splits the real line into several equiprobable pieces for the true mean. This provides a fully data-driven approach to construct the upper confidence bound, and this manuscript shows that the resulting algorithm nearly achieves the optimal regret when the noise distribution is known. Experiments are also provided.

**Strengths:**

Overall I like this paper. This paper asks a clean question and provides a satisfactory answer to it. Although the main observation comes from [Campi and Weyer, 2010], and the analysis of the algorithm essentially parallels the traditional UCB analysis, in my opinion the application to bandit problems is still nice and makes this paper interesting enough for a NeurIPS publication.

**Weaknesses:**

I don't see a huge weakness of this work, but I can list some minor ones:

1. The main novelty is the application of the observation in [Campi and Weyer, 2010] to a data-driven confidence bound in bandits, and everything else in the paper is pretty standard.

2. The requirement that the distribution is symmetric around its mean is pretty restrictive.

3. The computational complexity of the resulting algorithm could be high and may not lead to a practically useful algorithm.

Despite the above concerns, I'd like to reiterate that the merits weigh more than the weaknesses in my opinion, and I still like this paper.

Detailed comments:

Table I: I think the probabilities on the first two rows should be swapped.

Section 1: In my opinion, Section 1 should be a standalone section without resorting to bandits. In particular, the subscripts i should be removed.

Theorem 2, or the case 2^T < M: is the modification in Table I really necessary for the application to MAB? After inspecting the proof, I feel that simply choosing UCB(T, delta) = \infty whenever T < log_2(1/delta) also gives the same regret upper bound; is that true?


**Questions:**

1. Could the authors say anything about adaptivity (possibility or impossibility) when the reward distribution is not symmetric around the mean?

2. Theorem 3 still exhibits a gap compared with UCB with known noise distribution when Delta is large. Do you think this gap is essential (unavoidable for any algorithms), or can be closed using some other approaches?

---

> ### Author Rebuttal · Authors · 2023-08-08
>
> We thank the reviewer for insightful comments and for recognizing the utilization [Campi and Weyer, 2010] in the context of the multi-armed bandit problem.
>
> **@R1-A1) MARS performance for non-symmetric reward distributions**
>
> A new simulation was added to Supplementary material of the paper section 5 to demonstrate MARS’s performance when reward distribution is non-symmetrical. MARS was implemented when rewards have Bernoulli distribution, and the result was shown in Figure 3 supplementary material (in PDF file attached). The following text was added to the Section 5 of Supplementary material explaining the new simulation.
>
> *“To evaluate the robustness of MARS and its effectiveness where the assumption of symmetric reward distributions is not met, MARS is implemented on Bernoulli setup when the number of arms is $K = 2$ and the means are $\mu_1=0.5$ and $\mu_2=0.01$.
> The findings are depicted in Figure 3 (Please refer to  (Figure 3 Supplement material) in the attached PDF file). The results indicate although the setup does not meet the symmetric assumption, yet MARS outperforms Vanilla UCB and Thompson sampling in this setup. Its performance is also comparable to that of BESA.”*
>
> Additionally, the paper [\*] assesses the impact of asymmetric distributions on confidence regions for model parameters using similar approach. It shows that the confidence regions remain robust to small asymmetries, showcasing the method's reliability in such scenarios.
>
> [\*] Care, Algo, Balázs Csanád Csáji, and Marco C. Campi. "Sign-Perturbed Sums (SPS) with asymmetric noise: Robustness analysis and robustification techniques." 2016 IEEE 55th Conference on Decision and Control (CDC). IEEE, 2016.
>
> The following sentence is added to the future work below line 233 to mention above reference.
>
> *“Another interesting future direction is evaluation of the method on asymmetric rewads and robustification of approach following the same principles as proposed in the paper [\*].”*
>
> **@R1-A2) Computational Complexity of MARS**
>
> Computational complexity of approach depends on the choice of $\delta$ as MARS updates $\lceil 1/\delta \rceil$ sub-sampled means in each round for the chosen arm. In order to address this important aspect of MARS, we added the following content after line 162
>
> *“MARS necessitates keeping $\lceil 1/\delta \rceil$ sub-sampled means for each arm. This leads to a memory requirement of $O(Kn)$ when $\delta=1/n$. The computational complexity of MARS is also depends on the choice of $\delta$, as updating $\lceil 1/\delta \rceil$ sub-sampled means is performed in each round. Notably, to reduce the computational burden, the required Bernoulli variables in the algorithm can be pre-generated and stored before the start of the game.”*
>
> To further exploration of the efficiency of MARS and its applicability, we conducted an analysis of its runtime with alternative approaches. The relevant table can be found in the attached PDF, which has been added as an addition to Section 5 of the supplementary material.The following text also explain and analyse the result of the table.
>
> *“Table 1 displays the average runtime of different approaches for the multi-armed bandit with a Uniform setup when the number of rounds is set to $2000$. In MARS, the parameter $\delta$ is set to $1/1000$, requiring updates of $1000$ subsampled means in each round. As a result, it takes more time compared to Vanilla UCB, Thompson Sampling, and PHE. However, MARS is faster than approaches like GIRO, which store the full memory of rewards.”*
>
> **@R1-A3) Probabilities in the first two rows need to be swapped**
>
> The probabilities were swapped, and the typo was fixed.
>
> **@R1-A4) Section 1 can be standalone without bandits**
>
> As suggested, this section can be treated as a standalone section introducing a new upper confidence bound for random variables. The section has been modified, and the subscripts "i" have been removed.
>
> **@R1-A5) By simply setting $\text{UCB}(T, \delta) = \infty$ whenever $T < \text{log}_2(1/\delta)$, the same regret upper bound is achieved**
>
> By doing so, equation (3) in the proof of Theorem 3 in supplementary material is not true anymore as we use:
>
> $$\mathbb{P}\left(\mu_1\geq{\text{UCB}}_1(s,\delta)\right)=\delta$$
>
> Hence the modification is necessary.
>
> **@R1-A6) Does Theorem 3's gap with UCB under a known noise distribution for large Delta remain unavoidable, or can it be closed using alternative approaches**
>
> In the current version of the algorithm, we believe this gap is unavoidable, primarily due to the assignment of infinity to UCB in the initial iterations. It is notable that the mentioned modification and assignment enables obtaining a guaranteed confidence bound for any number of observations without directly using $\psi(\cdot)$ function or making any other distributional assumptions on arms.
>
> We would like to thank once again for your insightful feedback.

---

> > ### Comment · Reviewer_sKFK · 2023-08-16
> >
> > Thank you for the detailed response. I'll keep my score and continue to recommend acceptance.
> >
> > Regarding A5), I believe in the analysis of UCB you only need the inequality $\mathbb{P}(\mu_1 \ge \text{UCB}_1(s,\delta)) \le \delta$ right? So using $\text{UCB}_1(s,\delta) = +\infty$ for small $s$ does not hurt the order of the regret. Or equivalently these could be treated as forced exploration rounds.

---

### Author Rebuttal · Authors · 2023-08-08

We thank the reviewers for their valuable and generally positive feedback. We are encouraged that the reviewers found our work novel in terms of *“used a cute observation in [8] [...] provides a fully data-driven approach to construct the upper confidence bound, and this manuscript shows that the resulting algorithm nearly achieves the optimal regret [...]”* (**R1**), *“Rigorous theoretical analysis”* (**R2**), *“[...] quite interesting and seems to be significantly simpler than existing studies on data-dependent bounds”* (**R3**), *“Interestingly, these guarantees are achieved while the algorithm does not use the function  directly, hence avoiding sub-optimal choices of  [...]”* (**R4**), and *“Recently, the literature on non-parametric algorithms based on bootstrap and sub-sampling has proposed interesting new approaches, and this paper is in my opinion a nice addition to this literature, showing new potentialities”* (**R4**).

Before discussing the main changes and added empirical evaluations, it worthwhile to mention a few points about rebuttals.

+ The numbered citations refer to references in the submitted paper. The additional citations are labelled by [\*].

+ As two new algorithms were added to all simulations and Figure 2 (a), Figure 2 (b), and Figure 3 in the paper and Figure 1, 2 in the supplementary material were replaced with the new figures found in the attached PDF file. One new figure and table was added to the paper, both of which available in the provided PDF. The figures referenced in all our responses to reviews are the new figures in the attached PDF file.

+ To facilitate reference and prevent redundancy, we've labelled responses to reviewers as **@R{1/2/3/4}-A{1/2/…}**, e.g. **@R2-A5** mean Answer 5 in the rebuttal letter for Reviewer 2.

The principal modifications and enhancements made to the paper are outlined as follows:

**Improvement of empirical evaluation of approach**

As suggested by **@R3** two approaches, namely GIRO and PHE were added to all simulations and the following text was added to the paper after line 200:

*“In GIRO, the parameter $a$, which represents pseudo-rewards per unit of history, is set to $1$. Due to the high sensitivity of the PHE approach to the choice of the tunable parameter $a$, simulations were performed for two values of $a$.*

Additional simulations demonstrate the strength of MARS as a non-parametric and scale-free algorithm, exhibiting strong performance without relying on tail information. The PHE algorithm relies on an adjustable parameter. PHE with two values for that parameter was added to experiments. Interestingly, the parameter that performed remarkably well in the Gaussian setup exhibited the weakest performance in the uniform setup, and conversely, the parameter that excelled in the uniform setup displayed the poorest performance in the Gaussian setup. This contrast highlight the importance of scale-free methods like MARS in real world problems when reward distribution is unknown.

For further discussion on the added simulations please see the Review 3’s rebuttal under the heading **@R3-A1** for simulations added to the main paper and **@R3-A2** for simulations added to the supplementary material.

**Discussion on memory and computational complexity of algorithm and a simulation comparing runtime of algorithms**

To provide clarity on this significant algorithmic aspect, an additional paragraph was included in the paper, accessible at **@R1-A2**. Furthermore, a comparison of the MARS algorithm's runtime with that of other algorithms was included, and the outcomes are presented in the Table in the attached PDF. The simulations indicate that MARS is faster than GIRO, which retains the entire history of rewards. For a more detailed explanations on the runtime comparisons, please refer to **@R1-A2**.

**Revision of Literature Review and Clearer Articulation of Contributions**

As advised by **@R4** we revised the related work section to provide a more precise explanations of existing algorithms along with their respective limitations and strengths.  The main contribution of the paper which is *“MARS achieves logarithmic regret without using the function $\psi(\cdot)$. Hence it avoids sub-optimal choices of $\psi(\cdot)$”* was explained more in the section contribution. Please see **@R4-A1** for further details.

**MARS performance for non-symmetric reward distributions**

A new simulation was added to Supplementary material of the paper demonstrating MARS’s performance when reward distribution is Bernoulli (non-symmetrical). The result was shown in Figure 3 supplementary material in PDF file attached. The results indicate although the setup does not meet the symmetric assumption, yet MARS outperforms Vanilla UCB and Thompson sampling in this setup. Its performance is also comparable to that of BESA.

Moreover, robustification of the approach for asymmetric rewards was added as a topic of future research inspired by the paper [*].

[*] Care, Algo, Balázs Csanád Csáji, and Marco C. Campi. "Sign-Perturbed Sums (SPS) with asymmetric noise: Robustness analysis and robustification techniques." 2016 IEEE 55th Conference on Decision and Control (CDC). IEEE, 2016.
Please see **@R1-A1** for further details.

We would like to express our gratitude once more for the time and effort.

---

### Decision · Program_Chairs · 2023-09-21

**Decision:**

Accept (poster)

**Comment:**

This paper provides a novel nonparametric and parameter-free algorithm for the multi-armed bandit problem. The reviewers agreed that the algorithm is novel. On the other hand, the symmetric distribution assumption is strong. The empirical evidence against this weakness is quite limiting; I wonder if it is possible to find a pathological example where it clearly suffer a linear regret.

That said, the strength outweigh the weakness, thus recommending an accept.